# Impact of Multiple Factors on the Incidence of Developmental Dysplasia of the Hip: Risk Assessment Tool

**DOI:** 10.3390/medicina58091158

**Published:** 2022-08-25

**Authors:** Łukasz Pulik, Katarzyna Płoszka, Krzysztof Romaniuk, Aleksandra Sibilska, Andrzej Jedynak, Ignacy Tołwiński, Paulina Kumięga, Paweł Wojtyński, Paweł Łęgosz

**Affiliations:** 1Department of Orthopaedics and Traumatology, Medical University of Warsaw, 02-005 Warsaw, Poland; 2Department of Biostatistics and Translational Medicine, Medical University of Lodz, 92-215 Lodz, Poland; 3Department of Medicine, Medical University of Warsaw, 02-091 Warsaw, Poland

**Keywords:** DDH, screening, prevalence, hip ultrasound, risk calculator

## Abstract

*Background and Objectives*: Developmental dysplasia of the hip (DDH) is one of the most common musculoskeletal conditions in children. If not treated, it leads to disability, gait abnormalities, limb shortening, and chronic pain. Our study aims to determine the impact of multiple risk factors on the incidence of DDH and to develop an interactive risk assessment tool. *Materials and Methods*: We conducted a retrospective cohort study in the Outpatient Clinic for Children of the Medical University of Warsaw Hospital. The Graf classification system was used for universal ultrasonographic screening. In total, 3102 infants met the eligibility criteria. *Results*: The incidence of DDH in the study group was 4.45%. The incidence of DDH in the Warsaw population, Poland, during the study period was 3.73 to 5.17 (95% CI). According to the multivariate analysis, the risk factors for DDH were birth weight (OR = 2.17 (1.41–3.32)), week of delivery (OR = 1.18 (1.00–1.37)), female sex (OR = 8.16 (4.86–13.71)), breech presentation (OR = 5.92 (3.37–10.40)), physical signs of DDH (25.28 (8.77–72.83)) and positive family history in siblings (5.74 (2.68–12.31)). Our results support the recent hypothesis that preterm infants (<37 weeks) have a lower rate of DDH. *Conclusions*: A multivariate logistic regression predictive model was used to build the risk calculator. The DDH risk calculator will be evaluated in a prospective validation study.

## 1. Introduction

Developmental dysplasia of the hip (DDH) is one of the most common musculoskeletal disorders in children. DDH is an abnormal growth of the hip joint and surrounding tissues. It refers to a heterogeneous spectrum of abnormalities that range from mild acetabular defect to subluxation or complete dislocation of the femoral head [1]. Undetected and untreated DDH can cause severe disability, gait abnormalities, limb shortening, reduced range of motion in affected joints, and chronic pain [2]. Still, it is the leading cause of osteoarthritis and the main indication for total hip replacement in young adults [3]. Then the surgery is challenging and often requires dedicated techniques. In DDH, it is recommended to use a small diameter cup in a medialized position within the primary acetabulum; this ensures restoration of the rotation center and adequate head coverage [4]. However, it requires a thin liner, which in young patients can be associated with increased material wear and the risk of revision [5]. Therefore, bearing surfaces should be carefully chosen; Jamari et al. recommend the use of a Ti6Al4V head that can reduce contact pressure and polyethylene (PE) wear compared with CrMoCo [6].

The pathogenesis of DDH is unclear. However, the literature identifies several well-confirmed risk factors, such as female sex, left side, breech position, family history of DDH, first birth [7] and clicking of the hips on clinical examination [8]. The other risk factors studied are oligohydramnios, fetal macrosomia, multiple pregnancy (MuP), hyperlaxity, torticollis, clubfoot, metatarsus varus [9], and alterations in vitamin D level in children [10]. Genes, such as *PAPPA2*, *IL-6*, *COL2A1*, *HOXD9*, *GDF-5*, and *TGFB1* may be involved in DDH pathogenesis [11,12,13].

The diagnosis of DDH in newborns is based on clinical and ultrasound examination. The most widely used ultrasound methods were introduced by Graf, Harcke, and Terjesen [14]. Physical examination should include the leg length discrepancy test (Galeazzi test), stability test (Barlow and Ortolani tests), and detection of limited abduction [15]. However, even in experienced hands, the findings of physical examination in DDH may be subtle [16]. Most dysplastic hips diagnosed with ultrasound are normal on physical examination [17].

Currently, there are two primary approaches to DDH ultrasound screening, universal and selective. Ultrasound screening was introduced in several European countries, such as Austria (1991), Switzerland (1995), Germany (1996), and the Czech Republic [18]. As the first among Asian countries, Mongolia (2017) recently launched universal ultrasound screening [19]. A second screening plan established in the United States and England is based on a newborn clinical examination. In selective screening, hip ultrasound is recommended only in infants with positive findings on physical examination [15,20]. The first ultrasound is generally recommended before 6 weeks of life in universal screening countries and in the first week of life in cases with risk factors for DDH or positive physical examination [14]. It is a “custom to perform universal hip screening with ultrasonography” in Poland [21]. However, there are no official guidelines or recommendations. Visit time frames vary between cities and hospitals in Poland, and the first ultrasound is performed between 1 and 12 weeks of life [21,22,23].

The frequency of DDH depends on ethnicity, race, age of the studied population, diagnostic criteria, and screening method (physical examination, plain radiographs, ultrasound technique). The incidence of DDH can range from 0.1% in Africans to 7.6% in Native Americans [24]. The current incidence of DDH and risk factors at the first screening visit in Polish outpatient clinics remain unknown. Furthermore, little is known about the cumulative effect of the risk factors mentioned above on the incidence of DDH in the population.

This study aimed to determine the incidence of DDH in newborns in Poland using the Graf method. We also wanted to evaluate the possible risk factors for DDH in the population studied. Furthermore, our study presents a novel multivariate regression model for DDH and the first dedicated risk calculator for the population with universal ultrasound screening. In the Discussion section, we contrast the results of our study with other findings in the field of DDH.

## 2. Materials and Methods

The study design is a retrospective cohort study. The study setting was the Outpatient Clinic for Children of Orthopaedics and Traumatology Department of the Medical University of Warsaw, Poland. We examined all patients for eligibility in a specific time frame. The sample size was determined from the available literature. We adopted the sample size from the study by Roposch et al., who enrolled 1953 patients to construct the risk calculator in the selectively screened population [25]. The screening method used for each patient was the Graf classification system (I–IV). Certified medical assistants performed data collection on paper and in electronic form. Physical examination included hip orthopedic examination and the general examination. The consent to access and retrieve medical data from the archive of the Outpatient Clinic was obtained from the head of the Department of Orthopaedics and Traumatology of the Medical University of Warsaw, Poland. Due to the retrospective nature of the study, informed consent was not required. The Institutional Review Board of the Medical University of Warsaw approved the study protocol on 10 June 2019 (AKBE/227/2019). For this report, we used the STROBE statement for observational studies.

### 2.1. Study Participants

The study included all patients who attended the Outpatient Clinic for ultrasound screening from January 2013 to December 2018. Exclusion criteria were newborns who attended the first hip ultrasound screening visit in another facility and cases with missing information on the risk factors investigated. The population represents patients in the urban area. The ethnicity is relatively homogeneous. The number of live births in the catchment area (Warsaw, Poland) during the study period was 121,425 (2013—18,438, 2014—19,511 [25], 2015—19,905, 2016—20,980, 2017—21,315, 2018—21,276 [26,27]). The flow chart of included and excluded patients and reasons for exclusions is provided in Figure 1.

### 2.2. Variables

According to Graf’s classification, the diagnostic criteria for DDH were type IIa (-), IIb, IIc, D, III, and IV images. Type IIa hips before 6 weeks of age were monitored and treated only in the absence of signs of sufficient maturation (IIa (-) or IIb) [28]. Figure 2 presents ultrasound images from the data set with the measurements outlined. Our clinic recommends the first ultrasound examination at 6 weeks of life. In the case of a positive physical examination at birth or risk factors, ultrasound is recommended in the first weeks of life. The second control visit is also recommended for healthy children at 12 weeks. The history of hip orthopedic examination included the maximum abduction angle value for each hip joint, the Ortolani test, the Barlow test, and the Galeazzi test. The abduction asymmetry was defined as a difference of 20° or more. Articular noises on physical examination such as “clicks” or “creaks” were not classified as pathological findings [29]. The record included the name, national identification code, age, and date of visit. Information on possible risk factors such as female sex, abnormal presentation, high birth weight, term of birth, MuP, mode of delivery, diabetes, positive family history and coexisting medical conditions in children was collected.

### 2.3. Data Sources

The physical examination was conducted by an experienced orthopedic surgeon (PW, PG, WW, RW, and GT) who also performed ultrasound and the α and β angles measurements. An ultrasound device (Sonoline SI-400, Siemens, Berlin, Germany) operating with a 7.5 MHz linear transducer and a holding cradle was used. The diagnostic criteria for DDH were according to the Graf classification [28]. There was no limit on the age of a child for a hip ultrasound. The analysis included the first control visit to the clinic for type Ia, type Ib Graf hips. In patients with IIa at the first visit, the analysis included subsequent visits to assess whether the treatment was implemented in patients with insufficient hip maturation at the next visits.

### 2.4. Bias

Our study is limited to children who do not have obvious dislocations diagnosed at birth. This group of newborns is usually directed to dedicated centers specialized in pediatric orthopedic surgery. This fact could modify the study results and possibly lower the DDH rate in our cohort, especially Graf type III and IV with hip instability on physical examination and could be the source of selection bias. Thus, the ‘examination’ as a risk factor should be valued very carefully. The potential source of observation bias was that the ultrasound assessment was performed by the orthopedic surgeon examiner (PW, PG, WW, RW, and GT)**,** who also performed a physical examination and knew the DDH risk factors examined.

### 2.5. Quantitative Variables

We decided to adopt a cut-off value for limited hip abduction from Jari et al. [30], defined as a difference of 20° or more between both hips. We do not analyze the bilateral limitation of abduction. According to Jari et al [30]., bilateral limitation of hip abduction is not a useful indicator of DDH. However, the unilateral limitation of 20° or more abduction is a specific and sensitive sign of DDH. We adopted the definition of fetal macrosomia from the American College of Obstetricians and Gynecologists (ACOG) as birth weight greater than 4000 g regardless of gestational age [31]. Preterm delivery was defined as the birth before 37 weeks of gestation. We used the ACOG definition of postterm pregnancy (≥42 weeks) [32].

### 2.6. Statistical Methods

To summarize the characteristics of the studied groups, for continuous variables, the median was used together with the lower and upper quartiles (Q1–Q3) for continuous variables with a distribution other than normal. The normality of the distribution was verified with the Shapiro–Wilk test at a significance level of *p* < 0.05. Analysis of the differences between groups for nominal variables was carried out using 2 × 2 tables, the significance of which was verified by Chi2, Chi2 with Yates correction, or Fisher’s exact test, as appropriate. The results were reported with a *p*-value. A univariate logistic regression model was performed to verify the significant predictors for the analysis. A multivariate logistic regression was used to further evaluate predictors and their interactions, building a model using the backward stepwise method with the predictor cut-off point at *p* < 0.1. The odds ratio (OR) with 95% confidence intervals (95% CI) was calculated for each predictor. The diagnostic value of the model was evaluated by analyzing the area under the curve (AUC) of the Receiver operating characteristic (ROC) curve with its 95% confidence interval. Sensitivity and specificity were also calculated for the diagnostic test. The statistical significance for the analyses was set at *p* < 0.05. We used Statistica 13.3 analytics software from TIBCO (Palo Alto, CA, USA). The conditional probability calculator for multivariate logistic regression was calculated from the logistic regression formula. The variables for the presence of the factor were coded as 1—the presence of the risk factor DDH, 0—the lack of the risk factor DDH. The sex was coded as 1 for women, 0 for men. The weight in the model was determined in kilograms. The calculator was created using Excel Microsoft Office 2021.

## 3. Results

Among 4891 infants who underwent hip ultrasound in the Outpatient Clinic from 1 January 2013 to 31 December 2018, 3102 met the criteria (*n* = 6204 hips). We excluded ten initially screened infants who started care in an external facility. Subsequently, we excluded cases with missing information for the investigated risk factors. The flow chart of included and excluded patients and reasons for exclusions are provided in Figure 2.

### 3.1. Outcome & Descriptive Data

Females constituted 49.7% of participants, while males 50.3%. The median time of delivery was 39.00 weeks, and the median birth weight was 3.40 kg (Table 1). The mean time to the first visit from birth was 7.98 weeks (median 8 weeks, minimum 1 week, maximum 31 weeks). The second visit time for treated children was 11.43 weeks (median 12 weeks, minimum 7 weeks, and maximum 14 weeks) and for all children who attended the visit was 13.7 weeks (median 14 weeks, minimum 3 weeks, and maximum 21 weeks). The distribution of risk factors for DDH is presented in Table 2.

### 3.2. Main Results

The incidence of DDH in the population of Warsaw, Poland, during the study period was 4.45% (3.73–5.17; 95% CI). Bilateral DDH occurred in 60 of 138 cases of DDH (43.48%). The distribution of Graf hip types is presented in Table 3. Statistical significance was obtained for the factors: female sex (*p* < 0.001), breech position (*p* < 0.001), cesarean section (*p* = 0.035), positive family history of DDH in at least one parent (*p* = 0.027), positive family history of DDH in at least one sibling (*p* < 0.001), DDH physical signs (*p* < 0.001), and preterm delivery < 37 weeks (*p* = 0.003). In our study, 97.11% of DDH patients had at least one confirmed risk factor (female sex, cesarean section, breech position, family history of DDH). Only four patients with DDH had no significant risk factor (2.89%). In comparison, 68.79% of patients without DDH also had at least one confirmed risk factor. After exclusion of the female sex, 57.97% of patients with DDH had at least one risk factor. Treatment methods used where: Tübinger orthosis (*n* = 100; 72.46%), padded abduction diapers (*n* = 32; 23.19%), Frejka pillow (*n* = 2; 1.45%), Koszla abduction brace (*n* = 2; 1.45%) and Pavlik harness (*n* = 1; 0.72%). Parents of one child refused treatment (0.03%). Treatment was started on the first visit in 131 patients (94.93%) and the second visit in 7 patients (5.07%). Of 3102 patients in the study group, 2505 patients attended the second visit (80.75%).

### 3.3. Logistic Regression Model

Out of 15 variables, only 9 were statistically significant in the univariate model for the classification of the occurrence of DDH. The model classified as statistically significant predictors of DDH: female sex, cesarean delivery, breech position, delivery before 37 weeks, positive history of DDH in parents and siblings, and physical signs (Table 4). However, it should be noted that delivery < 37 weeks of gestation is a protective factor (OR = 0.18 (0.04–0.72)). In addition, continuous variables such as weight and week of delivery also, according to the model, statistically significantly increase the probability of DDH (weight: OR = 1.69 (1.22–2.34), week of delivery: OR = 1.22 (1.08–1.38)). In the multivariate model (Table 5), only the variables such as weight, delivery week, female sex, breech position, physical signs, and a positive history of sibling dysplasia had a statistically significant influence on the chance of developing DDH. The Hosmer–Lemeshow goodness of fit for the logistic regression test was *p* < 0.0001, indicating a poor model fit to the data. The area under the ROC curve (Figure 3) for the created predictive model was AUC-0.81 with a 95% confidence interval of 0.77–0.85, a sensitivity of 76.09%, and a specificity of 72.27%.

### 3.4. The Risk Calculator

A multivariate model was used to construct the DDH risk calculator in a Microsoft Excel chart (Appendix A). The conditional probability and OR of DDH can be obtained from the tool by selecting “yes” or “no” for binominal variables (female sex, breech presentation, physical signs, positive family history—siblings) and giving the exact value for continuous measures (weight, week of delivery).

## 4. Discussion

### 4.1. Key Results

The overall aim of the study was to assess the incidence of DDH in the Polish population and investigate whether the risk factors described in the literature for DDH are also reflected in this group of patients. The incidence of DDH in the study group was high (4.45%). Our work confirmed the already known risk factors for DDH (Table 2). We also discuss some other risk factors that have appeared in the literature. We constructed the DDH risk calculator, which can be used as a clinical decision tool in the future but needs external validation in a prospective study.

### 4.2. Interpretation

In Poland, before the implementation of ultrasonography, the DDH rate was relatively high, 6.80%, and hip dislocation was reported in 1.06% of the population [32]. The high DDH rate in this study was due to different diagnosis methods at the time and only suspected infants had undergone the diagnostic process. Using ultrasound screening, DDH was diagnosed in 5.60% of newborns in the first week of life (Synder et al., 2003 Łódź, Poland) [22]. In our study, the incidence of DDH in the university hospital in the capital city (Warsaw, Poland) during the first screening visit was lower (4.45%) than in the aforementioned study. This difference is probably due to the hip maturation curve and visit timing of 1 week vs. the median of 8 weeks. Diagnosis of DDH can depend on the time of the examination and the method used. As a child grows older, the hip joint matures, which can be observed with both ultrasound (α angle) [28] and radiographs (acetabular index, acetabular depth ratio) [33].

Our work confirmed some of the already known risk factors such as female sex, breech position, cesarean section, and a positive family history of DDH. However, according to the literature, there was not even one risk factor in up to 60–95% [34,35] of DDH cases (except sex). Furthermore, most children with risk factors do not develop DDH, and the disease can be observed only in 1–10% of cases. In contrast to these results, in our study, 97.11% of DDH had at least one statistically significant risk factor. However, in previous studies, female sex was not included [35,36,37,38]. After exclusion of sex as a risk factor, 57.97% of patients with DDH had at least one risk factor.

The female sex is considered one of the most important risk factors for DDH [36,39]. This strong relationship was also confirmed in our study OR = 8.16 (4.86–13.71). The mechanism of this connection is still under investigation [40]. Various theories explain it; the most common is the gender-dependent influence of hormones, particularly relaxin, on hip joint development [41]. It has an inhibitory effect on uterine muscle contractions and relaxes pubic symphysis. The role of relaxin, present in the blood serum, ultimately stimulates collagen turnover by increasing the secretion of matrix metalloproteinases (MMPs), collagenase, and a plasminogen activator [36]. In women, a higher expression of relaxin receptors was identified in the placenta [42] and the anterior cruciate ligament [43]. Although the function of relaxin is already known, there is no evidence that this mechanism is exclusively responsible for the higher frequency of DDH in women [40,44]

There is also a group of so-called mechanical factors in which there is pressure on the hip joint during pregnancy by the walls of the uterus or by the delivery canal tract at birth. One of the mentioned factors is the abnormal position and presentation of the fetus. Many authors have already described breech positioning as a risk factor [45]. According to Chan et al., complete breech vaginal delivery (3% of all births) is linked to a 17-fold increased risk of DDH (OR = 17.15; CI 95% 2.79–22.99), while the breech presentation resolved by Caesarean section relates to a 10-fold increase in risk (OR = 10.03; CI 95% 8.58–11.72) [46]. These findings were confirmed in a meta-analysis (35,139 infants) by Panagiotopoulou et al. [47]. Therefore, breech positioning is probably a risk factor for DDH during pregnancy and birth when significant forces are applied to the hip joint [48]. Our results are consistent with those of the existing literature. We recorded 173 (5.6%) babies in the breech position, of whom 21 developed DDH (OR = 5.92 (3.37–10.40)).

All authors describe abnormalities on physical examination as a risk factor, which was also unquestionably demonstrated in our study. We considered the following conditions as abnormalities: positive Ortolani/Barlow/Galeazzi test and/or hip joint abduction asymmetry of 20 degrees or more and abduction of the joint of less than 45 degrees, which is consistent with the available evidence [30]. Some patients do not show abnormalities on clinical examination, but ultrasound reveals DDH. In our study, abnormalities on physical examination predisposed to DDH with odds of 29.48 (10.81–80.41).

The available scientific knowledge indicates that a positive family history is one of the most important risk factors for DDH [36]. According to the consensus of the Committee on Quality Improvement, Subcommittee on Developmental Dysplasia of the Hip, a risk of DDH was defined as 6% in cases of healthy parents and recognized DDH in siblings, 12% in cases of confirmed DDH of the mother/father and 36% in cases where DDH was recognized DDH in one parent and a brother/sister [49]. The correlation was also confirmed in our findings. The authors state that a higher risk can also be observed if the disease was recognized in a first-degree cousin of the child. In these cases, the risk is specified at a level of 1.7% [35].

Some studies investigate the mode of delivery as a potential risk factor for DDH, but the results are inconclusive [3]. In our study, a cesarean section in univariate analysis was associated with the chance of developing DDH in a child (OR—1.44 (1.02–2.02)). However, when the factor was introduced into the multivariate model, where the development of DDH was influenced by several factors simultaneously, the cesarean section was no longer significant and was rejected from the model. A cesarean section was indicated in cases with a breech position, which is a significant risk factor for DDH itself. We also examined MuP as a potential risk factor for DDH. We did not identify a direct correlation between MuP and DDH, which is consistent with the literature [50].

Some authors consider the presence of congenital diseases to be one of the risk factors for DDH. Congenital Muscular Torticollis (CMT) can be associated with an increased risk of DDH to a level of 17%. Significant differences in correlations were also observed by sex: a five-fold increase in hip joint dysplasia was observed in male newborns with coexisting CMT compared to female newborns with CMT [36]. Some publications indicate congenital foot deformities, i.e., talipes calcaneovalgus and metatarsus adductus as a possible risk factor for DDH [35]. This relationship was not confirmed among our group of patients.

In our study, preterm delivery (<37 weeks) decreased the chances of DDH (OR = 0.18 (0.04–0.72)) but only in the univariate model. The theory explaining this phenomenon is the shorter exposure to maternal hormones and the lack of mechanical problems with intrauterine leg movement. Similar results were obtained in the study by Lange et al. [51] and data from the Swedish Medical Birth Register [52].

The literature indicates that most dysplastic hips diagnosed by ultrasound are normal on clinical examination (71.63%) [17]. Only 9 out of 129 patients treated for DDH had a positive physical examination in our study. This was probably due to the strict criteria of 20° of the difference in abduction angle versus contralateral side and cohort characteristics in the first clinical screening with physical examination at birth. Neonates with positive Ortolani and Barlow signs are placed directly in dedicated wards after birth.

Other risk factors sometimes mentioned in scientific discussions appear to be statistically insignificant in most studies. We examined some of them (i.e., Apgar Score < 10, oligohydramnios, fetal macrosomia, parity, post-term pregnancy). Our results are consistent with the results worldwide (Appendix A).

Sahin F et al. highlighted that the calculation of the risk of DDH in a patient could be used as a decision tool for ultrasound screening [53]. Similarly, Woodacre et al. proposed to modify the UK screening program by calculating the risk for each child [54]. This calculator could be used as a decision tool for screening in the future to define the urgency of the visit as the risk of treatment failure is higher in older children [55]. To assess the impact of risk factors on the occurrence of DDH, Roposch et al., proposed a risk calculator based on an analysis of patients selectively screened in the British population. Female sex, family history, physical examination, and birthweight were considered. The model demonstrated excellent discrimination and calibration of the observed and predicted risk [56]. Compared to the Roposch et al. risk calculator in which only newborns subjected to selective ultrasound screening were enrolled, our model is based on universal ultrasound screening and includes all children regardless of signs or risk factors of DDH [56].

### 4.3. Limitations

The retrospective study influences data collection and increases the missing value rate. Furthermore, the results do not reflect the situation for the whole country. This single-center study covers only 4.03% of newborns in Warsaw, Poland, during the study period. In Warsaw, there are many clinics where the screening is performed and parents of patients can freely choose where the ultrasound examination is carried out. Therefore, additional well-designed multicenter prospective studies on this subject are required. The accuracy of the examination, especially with respect to specificity, is closely related to the examiner’s skills. The Graf method must be performed in strict compliance with the author’s instructions. Only certified orthopedic surgeons performed the ultrasound examination (PW, PG, WW, RW, and GT) [28]. The reported age at the first visit ranges from 1 to 31 weeks. Various factors can cause this substantial age difference. Some children are directed for an immediate hip ultrasound if they have symptoms of DDH or significant risk factors. On the other hand, some children present late because of illness or other unrelated reasons. Early hip ultrasound can cause a high rate of diagnosis of hip immaturity, which should not be treated and resolves spontaneously. However, only limited evidence shows that moderate/mild DDH can resolve spontaneously over time [14].

## 5. Conclusions

Taken together, these findings demonstrate that the DDH rate in Warsaw, Poland is high. Furthermore, we confirmed the risk factors for DDH: female sex, breech presentation, and positive family history of DDH in parents and siblings. Interestingly, the results reveal that preterm infants (<37 weeks) have a lower rate of DDH. This work has presented a novel method for DDH risk calculation to measure the cumulative effect of risk factors on DDH in the universally screened population. Further research is needed to evaluate this tool and its error rate and tolerance in a prospective study.

## Figures and Tables

**Figure 1 medicina-58-01158-f001:**
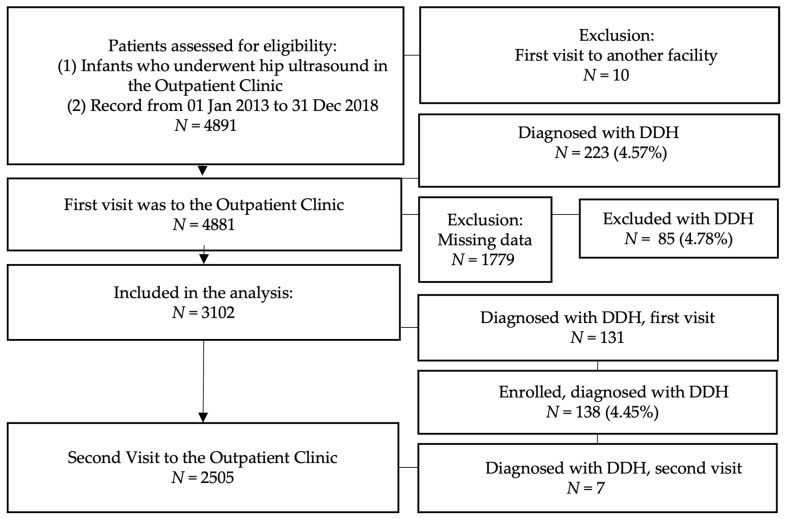
The flow chart of included and excluded patients and reasons for exclusions.

**Figure 2 medicina-58-01158-f002:**
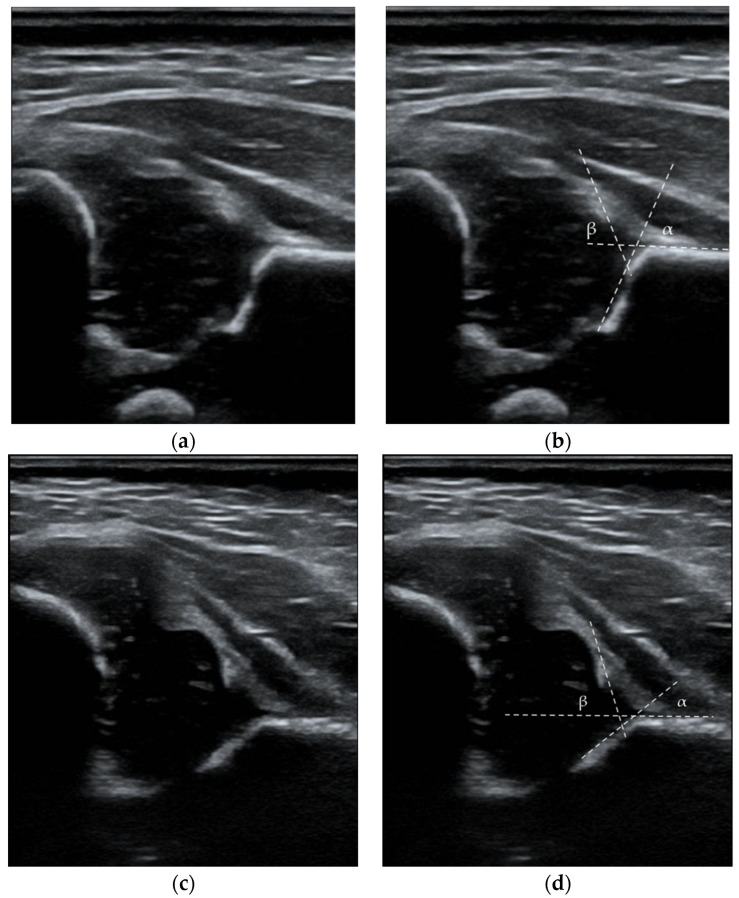
Ultrasound images with the measurements outlined: (**a**,**b**) Type IB, hip joint according to Graf, α = 68° β = 67°; (**c**,**d**) Type III, hip joint according to Graf, α = 42° β = 73°. White lines represent the base line, the bony roof line, and the cartilaginous roof line.

**Figure 3 medicina-58-01158-f003:**
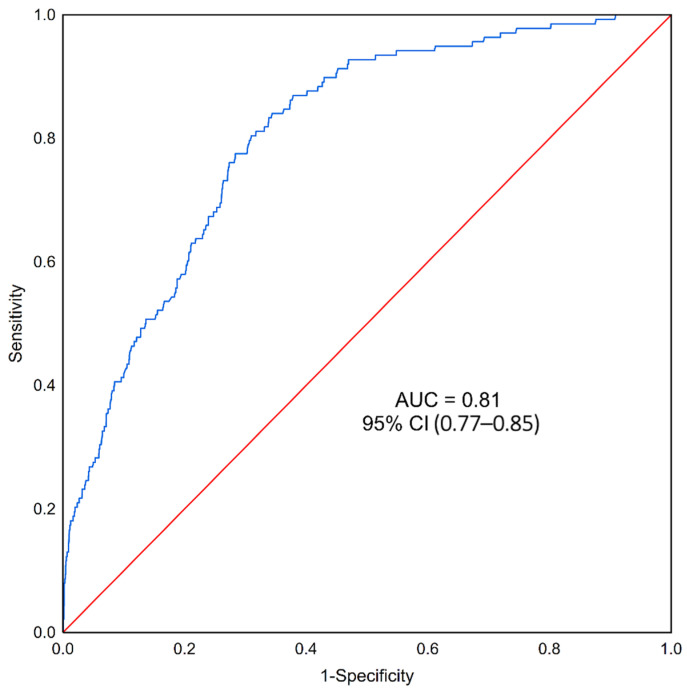
Multivariate logistic regression predictive model for developmental dysplasia of the hip (DDH) in children (Statistica 13.3). The blue line represents a receiver operating characteristic (ROC) curve.

**Table 1 medicina-58-01158-t001:** Demographic data of patients.

Variable	Median (Q1–Q3)
Time of delivery (week)	39.00 (38.00–40.00)
Birth weight (kg)	3.40 (3.09–3.73)
First visit (week)	8.00 (1.00–31.00)

**Table 2 medicina-58-01158-t002:** Risk factors of developmental dysplasia of the hip (DDH).

Variable	DDH—Yes	DDH—No	χ^2^
Sex			
Female	120 (7.8%)	1421 92.2%)	<0.001
Male	18 (1.1%)	1543 (98.9%)	
Cesarean section			
Yes	68 (5.4%)	1194 (94.6%)	0.035
No	70 (3.80%)	1170 (96.2%)	
Delivery presentation—Breech			
Yes	21 (12.1%)	152 (87.9%)	<0.001
No	117 (4.0%)	2812 (96.0%)	
Delivery < 37 weeks			
Yes	2 (0.9%)	228 (99.1%)	0.003
No	136 (4.7%)	2736 (95.3%)	
Positive family history of DDH—parents			
Yes	17 (7.3%)	215 (92.7%)	0.027
No	121 (4.2%)	2749 (95.8%)	
Positive family history of DDH—siblings			
Yes	10 (19.2%)	42 (80.8%)	<0.001
No	128 (4.2%)	2922 (95.8%)	
Physical signs			
Yes	9 (56.3%)	7 (43.7%)	<0.001
No	129 (4.2%)	2957 (95.8%)	

χ^2^—the significance of *p* for χ^2^.

**Table 3 medicina-58-01158-t003:** Hip type according to Graf in study participants.

Hip Type According to Graf		Right Hip	Left Hip
Type I	A	1297 (41.81%)	1326 (42.75%)
	B	1645 (53.03%)	1640 (52.87%)
Type II	A	123 (3.97%)	96 (3.09%)
	B	6 (0.19%)	6 (0.19%)
	C	19 (0.61%)	23 (0.74%)
Type III		11 (0.35%)	11 (0.35%)
Type IV		1 (0.03%)	-

**Table 4 medicina-58-01158-t004:** Univariate logistic regression to evaluate DDH predictors.

Variable	Effect Level	β	*p*	OR (95% CI)
Weight (kg)		0.52	0.002	1.69 (1.22–2.34)
Week of delivery		0.20	0.001	1.22 (1.08–1.38)
Sex	F/M	1.98	<0.001	7.24 (4.39–11.95)
Cesarean section	Yes/No	0.36	0.037	1.44 (1.02–2.02)
Multiple pregnancy	Yes/No	−0.78	0.279	0.46 (0.11–1.88)
Macrosomia > 4000 g	Yes/No	0.21	0.421	1.24 (0.74–2.09)
Delivery presentation—Breech	Yes/No	1.20	<0.001	3.32 (2.03–5.43)
Delivery ≥ 42 weeks	Yes/No	0.81	0.066	2.24 (0.95–5.28)
Delivery < 37 weeks	Yes/No	−1.73	0.015	0.18 (0.04–0.72)
First birth	Yes/No	−0.11	0.525	0.89 (0.63–1.26)
APGAR score < 10	Yes/No	−0.03	0.920	0.97 (0.57–1.66)
Positive family history of DDH—parents	Yes/No	0.59	0.029	1.80 (1.06–3.04)
Positive family history of DDH—siblings	Yes/No	1.69	<0.001	5.44 (2.67–11.08)
Oligohydramnios	Yes/No	−0.15	0.881	0.86 (0.12–6.38)
Physical signs	Yes/No	3.38	<0.001	29.48 (10.81–80.41)

*p*—significance; OR (95% CI)—Odds ratio with 95% confidence interval; F—female; M—male.

**Table 5 medicina-58-01158-t005:** Multivariate logistic regression model to predict the incidence of DDH in children.

Variable	Effect Level	β	*p*	OR (95% CI)
β_0_		−13,892	<0.0001	0.00 (0.00–0.00)
Weight (kg)		0.772	0.0004	2.17 (1.41–3.32)
Week of delivery		0.162	0.0381	1.18 (1.00–1.37)
Sex	F/M	2099	<0.0001	8.16 (4.86–13.71)
Breech presentation	Yes/No	1778	<0.0001	5.92 (3.37–10.40)
Physical signs	Yes/No	3229	<0.0001	25.28 (8.77–72.83)
DDH—siblings	Yes/No	1747	0.0001	5.74 (2.68–12.31)

*p*—significance; OR (95% CI)—Odds ratio with 95% confidence interval.

## Data Availability

The data sets analysed during the current study are available from the corresponding author on reasonable request.

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
