# Peer review of "Impact of Multiple Factors on the Incidence of Developmental Dysplasia of the Hip: Risk Assessment Tool"

_medicina, 2022, doi:10.3390/medicina58091158_

Round 1

Reviewer 1 Report (New Reviewer)

1.      Line 2-3, Uppercase and lowercase of the present title should be revised according to MDPI format.

2.      Email from all of the authors should be provided after affiliation with initials if in one affiliation if more than one email.

3.      Please use keywords that are mostly sown in the manuscript for searchable in the database. For example, “developmental dysplasia of the hip;” is never shown even once in the manuscript.

4.      What is the novel of the present study? Developmental dysplasia with risk assessment has been widely studied in the past. It is urgently needed to explain the novelty of more advances in the introduction section.

5.      The authors need to explain the previous research, their findings, novelty, and limitation to show the state of the art of the present study.

6.      For extending the explanation to become more comprehensive, the reviewer strongly encouraged the authors for explaining implant survival after hip replacement in a patient with developmental dysplasia in the introduction and/or discussion section. Also, to support this explanation, suggested references published by MDPI should be adopted to support this explanation as follows: Computational Contact Pressure Prediction of CoCrMo, SS 316L and Ti6Al4V Femoral Head against UHMWPE Acetabular Cup under Gait Cycle. J. Funct. Biomater. 2022, 13, 64. https://doi.org/10.3390/jfb13020064

7.      In the materials and methods, the authors need to add additional illustrations as a form of figure that explains the workflow of the present study to make the reader easier to understand rather than only the dominant text as a present form.

8.      What is the standard, basis, or reference that supports of explanation of the study participant? It is needed to be explained.

9.      More detail in tools information such as manufacturer, country, and specification needs to be stated.

10.   Error and tolerance of tools are important information that needs to be explained in the manuscript. Please provide it.

11.   Results comparison with similar previous studies needs to give.

12.   Please revise the conclusion as a section, not a subsection.

13.   Elaborate the conclusion as a form of paragraph, not point by point as present form.

14.   The authors need to enrich the reference from five years back. Literature published by MDPI is strongly recommended.

15.   The authors need to proofread the manuscript due to grammatical errors and language style.

Author Response

Thank you for your in-depth review of the manuscript. Your comments are valuable and very helpful in reviewing and improving the manuscript. We have revised the manuscript accordingly and provided specific responses below. The article has undergone extensive proofreading and language editing, as suggested. For the convenience of reviewers, the revisions have been marked with the track changes function. We look forward to hearing from you regarding our submission and responding to any further questions and comments you may have.

1. Line 2-3, Uppercase and lowercase of the present title should be revised according to MDPI format.

Response: We thank you for pointing this out. We have revised the title of the article according to MDPI format (line 2-3).

2. Email from all the authors should be provided after affiliation with initials if in one affiliation if more than one email.

Response: We provide the manuscript with emails from all authors accordingly (line 6-13).

3. Please use keywords that are mostly sown in the manuscript for searchable in the database. For example, “developmental dysplasia of the hip;” is never shown even once in the manuscript.

Response: First, we followed the rules of MeSH (Medical Subject Headings) systems for keywords, but as suggested, we changed the list of keywords to better reflect the content of the manuscript (line 28).

4. What is the novel of the present study? Developmental dysplasia with risk assessment has been widely studied in the past. It is urgently needed to explain the novelty of more advances in the introduction section.

Response: The novelty of the present study is explained now in the Introduction section. This is the only study of the current incidence of DDH in newborns in Poland based on universal ultrasound screening. Furthermore, our study presents a novel multivariate regression model for DDH and the first dedicated risk calculator for the population with universal ultrasound screening. It could be used to define the urgency of the visit as the risk of treatment failure is higher in older children. (line 79-84).

5. The authors need to explain the previous research, their findings, novelty, and limitation to show the state of the art of the present study.

Response: We discussed previous research and the novelty of the current study in the discussion section. Limitations of current study is now elaborated in the Limitations (4.3) subsection (line 409-426).

6. For extending the explanation to become more comprehensive, the reviewer strongly encouraged the authors for explaining implant survival after hip replacement in a patient with developmental dysplasia in the introduction and/or discussion section. Also, to support this explanation, suggested references published by MDPI should be adopted to support this explanation as follows: Computational Contact Pressure Prediction of CoCrMo, SS 316L and Ti6Al4V Femoral Head against UHMWPE Acetabular Cup under Gait Cycle. J. Funct. Biomater. 2022, 13, 64. https://doi.org/10.3390/jfb13020064

Response: We thank the reviewer for pointing this out. We have revised the Introduction section and cited the article by Jamari and colleagues, which indicates that the standard CoCrMo head in young patients (e.g., with DDH) may be inferior to other materials such as Ti6Al4V (line 43-44).

7. In the materials and methods, the authors need to add additional illustrations as a form of figure that explains the workflow of the present study to make the reader easier to understand rather than only the dominant text as a present form.

Response: We add Figure 1 explaining the workflow of the present study (line 141-143).

8. What is the standard, basis, or reference that supports of explanation of the study participant? It is needed to be explained.

Response: We have made the change. We adopted the sample size from the study by Roposch et al. who enrolled 1953 patients to construct the risk calculator in the selectively screened population.

9. More detail in tools information such as manufacturer, country, and specification needs to be stated.

Response: We thank the reviewer for pointing this out. We have revised the manuscript, and this information is now provided in the Data Sources (2.3) subsection (line 90-91).

10. Error and tolerance of tools are important information that needs to be explained in the manuscript. Please provide it.

Response: The predictive model used in the calculator was AUC-0.81 with a 95% confidence interval of 0.77-0.85, a sensitivity of 76.09%, and a specificity of 72.27%. We appreciate the insightful suggestion of the reviewer and agree that it would be useful to present the measurement properties of the calculator. However, such an analysis is beyond the scope of our article and will be performed in the prospective validation study, as stated in the Conclusions section (line 433-434).

11. Results comparison with similar previous studies needs to give.

Response: The topic is elaborated in the Discussion section.

12. Please revise the conclusion as a section, not a subsection.

Response: We have made the proposed change, and the conclusion is now the title of the section (line 426).

13. Elaborate the conclusion as a form of paragraph, not point by point as present form.

Response: We agree and have updated the conclusion section (line 428-434).

14. The authors need to enrich the reference from five years back. Literature published by MDPI is strongly recommended.

Response: Reviewed accordingly, now we cite the following literature published by MDPI:

Wang, H., et al., Asymmetry in Muscle Strength, Dynamic Balance, and Range of Motion in Adult Symptomatic Hip Dysplasia. Symmetry, 2022. 14(4): p. 748.

Zamborsky, R., et al., Developmental Dysplasia of Hip: Perspectives in Genetic Screening. Medical Sciences, 2019. 7(4): p. 59.

Harsanyi, S., et al., Genetic Study of IL6, GDF5 and PAPPA2 in Association with Developmental Dysplasia of the Hip. Genes, 2021. 12(7): p. 986.

 Janz, V., et al., Developmental Hip Dysplasia Treated with Cementless Total Hip Arthroplasty Using a Straight Stem and a Threaded Cup—A Concise Follow-Up, At a Mean of Twenty-Three Years. Journal of Clinical Medicine, 2021. 10(9): p. 1912.

 Jamari, J., et al., Computational Contact Pressure Prediction of CoCrMo, SS 316L and Ti6Al4V Femoral Head against UHMWPE Acetabular Cup under Gait Cycle. Journal of Functional Biomaterials, 2022. 13(2): p. 64.

15. The authors need to proofread the manuscript due to grammatical errors and language style.

Response: We sincerely appreciate the comments of the reviewer. The manuscript was proofread, and the language was corrected as suggested.

Reviewer 2 Report (New Reviewer)

Article: Impact of multiple factors on the incidence of developmental dysplasia of the hip: risk assessment tool.

Retrospective study of a cohort of neonates attended at an Outpatient Clinic for Children of Orthopedics and Traumatology Department of the Medical University of Warsaw, Poland.

It is a screening of children who do not present symptoms/examination compatible with DDH at birth (if they do present it, they are referred to other centres), classifying the cases according to the Graft ultrasound classification.

Once classified, a study of the risk factors is made, determining a univariate analysis, first, and a multivariate analysis later.

The study is correct, although not very novel.

It presents some aspects that I would like the authors to value:

- There is a significant percentage of cases excluded due to missing data (these “missing” cases should be assessed since they represent 36% of the sample.

- The fact that the cases with examination/clinical DDH at birth have been excluded means that “examination” should be valued very carefully among the risk factors.

- It would be interesting to clarify some details: are all cases evaluated twice, or only pathological ones? If not, it should be made clear how many cases make the second assessment. In the flow graph S1 the result of the first and the second review could be added.

- The risk calculator should be better explained. The supplementary material does not allow this risk to be calculated (8 is a pdf). This should be improved.

Formally, the manuscript is correct, but the following aspects should be assessed:

- Personally, I would remove at least one decimal from the weeks of gestation and the weeks of analysis.

- Change Gender for Sex: gender should not be used in the newborn, in my opinion, since the newborn is not yet aware of "gender" (in any case, this is a decision that should follow the editorial style book ). In the text “sex” is used but in the tables “gender” is used. You have to unify it.

Author Response

1. There is a significant percentage of cases excluded due to missing data (these “missing” cases should be assessed since they represent 36% of the sample.

Response: While we understand the reviewer's concern, we included the information on DDH incidence in patients with missing data (Figure 1, line 151).

2. The fact that the cases with examination/clinical DDH at birth have been excluded means that “examination” should be valued very carefully among the risk factors.

Response: We agree with your comment. We now include this information in the Bias Subsection (2.4) (line 169-170)

3. It would be interesting to clarify some details: are all cases evaluated twice, or only pathological ones? If not, it should be made clear how many cases make the second assessment. In the flow graph S1 the result of the first and the second review could be added.

Response: All patients are advised for evaluation before 6 weeks of life and in 12 weeks of life. We include the information in the flow graph (Figure 1).

4. The risk calculator should be better explained. The supplementary material does not allow this risk to be calculated (8 is a pdf). This should be improved.

Response: We thank the reviewer for pointing this out. We have made the change and the tool is now included in the Excel file that allows the calculation of cumulative risk. The use of the calculator is explained now in the subsection 3.4 (line 265-270).

Formally, the manuscript is correct, but the following aspects should be assessed:

5. Personally, I would remove at least one decimal from the weeks of gestation and the weeks of analysis.

Response: In the multivariate model, only a 'week of gestation' is included, which is a significant risk factor for DDH confirmed in other studies. This is now elaborated on in the discussion section. As suggested the weeks of analysis are excluded.

6. Change Gender for Sex: gender should not be used in the newborn, in my opinion, since the newborn is not yet aware of "gender" (in any case, this is a decision that should follow the editorial style book ). In the text “sex” is used but in the tables “gender” is used. You have to unify it.

Response: This observation is correct. We have changed ‘gender’ for ‘sex’ both in the tables and in the text as suggested.

Round 2

Reviewer 1 Report (New Reviewer)

The Reviewer believes it is suitable for publication in the present form.

This manuscript is a resubmission of an earlier submission. The following is a list of the peer review reports and author responses from that submission.

Round 1

Reviewer 1 Report

I thank the authors for the opportunity to review this paper which investigates the impact of multiple risk factors on the risk of DDH and develops a risk assessment tool based on retrospective data from a universal ultrasound screening program for DDH in Poland.

General comments. While generally well structured, the study would benefit from extensive proof-reading and English language editing. There are several instances of words missing, grammatical errors and confusing sentence structures.

Studies of risk factors related to DDH are numerous, but there are a limited number of studies in populations that are universally ultrasound screened, as is claimed in the present article, giving this study merit.

Section specific comments

Introduction:

-          The background section could be shortened and should focus on the subject examined and relevant work preceding the current article. The debate on selective vs universal ultrasound screening for DDH, while extremely relevant, is not the focus for the present article and should be omitted or significantly reduced.

-          I would suggest presenting studies concerning previously proposed risk factors such as the meta-analysis of Ortiz-Neira [44] and De Hundt, M. et al. Risk factors for developmental dysplasia of the hip: A meta-analysis. Eur. J. Obstet. Gynecol. Reprod. Biol. 165, 8–17 (2012).

-          The most relevant article to include in the background section is the recent similar study by Roposch et al [49]. This study is mentioned in the final part of the discussion but needs to be addressed in the introduction as it concerns the justification for the current study. Why redo what Roposch did? What sets the current study apart from the Roposch study?

-          In the discussion line 346 the authors claim that the main difference between the present study and the study by Roposch et al is the universal ultrasound screening at their institution. However, in line 70 the authors write “There is no official universal screening program or guidelines for the DDH screening and control visits timeframes in Poland. However, ultrasound examination is done in most children”. This needs to be clarified as a universally screened population is the main strength of the study, what does “most children” constitute? When is the screening done?  

-          In order to account for potential selection bias and to calculate the population incidence, the authors should present the birth rate for the catchment area for the institution during the inclusion period.

Materials and methods

-          Line 87-88: this is also explained in the Study participants section and should be omitted.

-          Line 89: “Study size was determined from the study type and included all who met eligibility criteria for participation.” How was the study size determined? Was a sample size calculation made? Please elaborate.

-          What were the inclusion/exclusion criteria for the study?

-          When was the diagnosis made? Line 109: “First ultrasound examination is recommended at 6 weeks of life. In case of a positive physical examination upon birth or risk factors, ultrasound is recommended upon first weeks of life”. The reported age at first visit ranges from 1 to 31 weeks. This substantial age difference is not commented on by the authors.

This is a potential major limitation, as we know that mild/moderate hip dysplasia resolves spontaneously with time and the incidence of DDH therefore changes with the age of the population examined.

o   The authors regard Graf type IIa(-) hips as DDH positive. Which implies that the child is at least 6 weeks old when this diagnosis is made. As the diagnosis was made at the first clinical visit and the mean age of the child at this visit ranged from 1 to 31 weeks, would this not introduce a selection bias in the primary outcome as some children could receive a type IIa(-) diagnosis where as others couldn’t as they were below 6 weeks of age at the first visit and consequently would be rated as a Graf type IIa+? How do the authors account for this?

-          Line 111: “The second control visit is also recommended for healthy children at 12 weeks” How many attended this follow up visit? How did the examined risk factors predict DDH at initial visit and 12 weeks follow-up?

-          Line 123: “The physical examination was conducted by an experienced orthopedic surgeon who also performed the ultrasound”. While unavoidable given the logistical setup of the authors’ institution, this a source of observation bias that should be commented on.

-          Line 126: “There was no age limitation for hip ultrasound. According to Graf the ultrasound should not be assessed only if the ossification nucleus of the femoral head blocks the signal from lower limb of the ilium”. The meaning is unclear to me, consider rephrasing. Are the authors suggesting that there is no limit on the age of a child for hip ultrasound?

-          It is generally helpful to include an image of a hip ultrasound with Alfa and beta angle annotations to confirm correct usage of the Graf method.

Statistics

-          How were the bilaterality in data handled?

-          A ROC curve for the constructed model was made. As I read the article, the model is based on dichotomous variables, is it possible to make a ROC curve for dichotomous predictors?

-          A significance level of 0.05 was applied. Given the number of factors analyzed for significant association to DDH, was a lowering of the significance value considered? (i.e. Bonferonni correction?)

Results

-          In order to determine the external validity of these results, the authors should give the proportions of Graf type hips included in this study, not only their dichotomous DDH yes/no variable.

-          Line 176: “Ten initially screened infants excluded from the study were continuing care started in an external facility”. I do not understand this, where they excluded and later diagnosed with DDH? Where the others normal? What were the exclusion criteria?

-          Present number of births in the catchment area of the institution for the inclusion period

-          Table 1: percentages are wrong for gender variable (total is 103.3%)

-          Table 1: should give row percentages instead of row/column percentages for comparison across exposure variables.

-          Line 193: “in our study, 97.11% of DDH patients had at least one confirmed risk factor”. This number seems very high given earlier reports that 85% of patients who received arthroplasty later in life because of DDH did not have a positive clinical examination or risk factor at birth.  Sink, E. L., Ricciardi, B. F., Torre, K. Dela & Price, C. T. Selective ultrasound screening is inadequate to identify patients who present with symptomatic adult acetabular dysplasia. J. Child. Orthop. 8, 451–455 (2014). I would suspect a selection bias playing a role here (i.e. parents of children with risk factors are more likely to show up for a hip ultrasound), how do the authors explain this high number?

-          Table 3: Weight/Week of delivery. How is the effect (0.52/0.20) interpreted? 0.52 per kg or week above a certain threshold? Please elaborate.

Discussion

-          The authors need to comment on the wide range of age at first clinical visit, and how it can impact the DDH incidence.

-          Line 253: “Our work confirmed some of the already known risk factors such as […] cesarean section.” Cesarean section was calculated with a p value of 0.03, see earlier comment regarding significance level, is this significant?

-          line 263-272: not relevant to the current study. Should be deleted.

-          Line 280:  Dezateux et al is a literature review. The authors should reference the original papers referenced by Dezateux et al, instead of the review paper.

-          Line 287: was the Galeazzi test not performed? Or was it not considered an abnormality?

-          Line 290: as the orthopaedic surgeon who performed the clinical examination also performed the ultrasound, the influence of observation bias needs to be mentioned.

-          Line 300-302: not relevant for the current study. Should be deleted.

-          Line 308: why was this? Confounding error?

-          Line 339: the negative predictive value of any test for a rare condition (DDH) is bound to be high, and not of large interest. See Mausner JS, Kramer S: Mausner and Bahn Epidemiology: An Introductory Text. Philadelphia, WB Saunders, 1985, p. 221

-          Line 342-347: This should be the central argument in the introduction.

-           

Limitations: See all the above, there are several limitations and biases to the current study, most notably the wide age range of examined children and the missing information on the proportion of children screened/born.

Generalizability

-          Missing, it seems that the authors instead have put their conclusions?

Conclusion:

-          “The cumulative effect of well-known risk factors can be assessed with a DDH risk calculator”

o   What constitutes well-known?

o   The risk calculator was not examined in this paper, thus the paper does not support this conclusion

Reviewer 2 Report

Manuscript review: Impact of multiple factors on the incidence of developmental dysplasia of the hip: risk assessment tool

This is a retrospective cohort study from a single institution investigating the risk factors for DDH I cohort of 3102 Infants. The Graf classification was used in a ultrasonographic universal screening program. The diagnostic criteria for DDH were type 2a (-) or above. The following risk factors were: female sex, abnormal presentation, birth weight, term of birth, multipara, mode of delivery, diabetes, positive family history and co-existing medical conditions. Univariate and multivariate logistic regression analysis were performed. Odd radios with 95% confidence intervals were calculated for each predictor. A conditional probability calculator constructed based on the multivariate logistic regression predictive model. Hereby a risk assessment tool was created. The study confirmed known risk factors for DDH and supported the recent findings that preterm infants have a lower rate of DDH.

The overall assessment is that this study confirms existing knowledge but contribute only to a minor degree with new knowledge. However, studies like this are still important for confirming existing knowledge and may serve as literature for later metanalyses. According to the title of the study the main outcome is the risk assessment tool. However, I miss in the discussion how this tool is going to be applied and it's difficult to see the value of the tool in a setting of universal ultrasounds The authors should argue why this tool is of value. Furthermore, I think the study would benefit from a statistical review.

More specific comments:

In the results section flow diagram of included and excluded patients and reasons for exclusions should be provided. Furthermore, a table with patient demographics would be preferable.

 In Table 1: Two decimals for percentages are given. This is unnecessarily detailed.

Page 3, line 118: MuP – use full wording first time

Page 3, line 124: An ultrasound image from the data set with the measurements outlined would be useful. Many interpretations on how to draw the correct lines exists.

Page 3, line 137: It is mentioned the skills of the examiner may influence the quality of the measurements, This is indeed true, but the will not bias the results unless fx all low risk patients were examined only by a certain group of examiners. However, it will influence the precision of measurements.

Page 5, line 187 it is stated that the incidence of DDH in the study Group that needed treatment was a 4.45%. This should be rephrased the number of patients who needed treatment is unknown. Instead, treatment was provided in 4.45% of the screened population. The indication for initiating treatment should be stated as well.

Page 5, line 193: It is stated that 97% of DDH patients had at least one confirmed risk factor. However, the number of patients with risks factors in the group without DDH needs to be given as well.

Page 8, line 255: The sentence starting with according to literature should be supported by references

Page 8, line 257: The authors mentioned that 97% of their patient with DDH at least one statistically significant risk factor and this is in contrast with the existing literature. This is indeed a surprising finding, and one may suspect that the cohort studied in the present study is based. The authors should discuss possible explanations for this.

Table 4: beta should explained.

In the discussion section the diagnostic criteria of DDH should should be discussed. Why did the authors choose to define Graf 2a(-) as the lower limit for DDH. 

Round 2

Reviewer 1 Report

Thank you for the revised manuscript.

At this time I will only focus on the justification of the study, which I find lacking.

Per my original comments, the main justification for this study is the alleged universal ultrasound screening for DDH. The authors have now presented the birth rate for the inclusion period for the catchment area. 121.425 children were born during the study period but only 4891 children (4%) were examined for eligibility for this study after having visited the Outpatient clinic for ultrasound screening.

Are these numbers correct?

The screening method may very well be universal ultrasound screening, but if only 4% of the population shows up for an ultrasound scan, this makes the inclusion rate much lower than even selective screening programmes.

This is a major limitation as it provides a clear source of selection bias which severely weakens the conclusions drawn from the results of the present study.

If the abovementioned numbers are correct, I do not recommend publication of this study as I do not have faith in the conclusion drawn from the population sample without compelling arguments to do so, which the authors have not provided.

Round 3

Reviewer 1 Report

I thank the authors for their clarifying comments and their explanation of the difficulties with reporting a response rate due to the logistical reality at their institution.
However, the central problem of the justification still stands in my opinion.

As the authors previously commented:

Compared to Roposch et al. risk calculator in which only newborns subjected to selective ultrasound screening were enrolled, our model is based on universal ultrasound screening and includes all children regardless of DDH signs or risk factors.

This study reports on a select subpopulation in a large urban area which are supposedly screened mainly on their own initiative. And as the authors have commented "There are numerous outpatient clinics and hospitals in Warsaw that perform hip ultrasound screening (some are also located downtown), so we do not cover all the population."

The percentage of newborns screened can therefore not be assessed, unless the authors could acquire the data from all the other screening clinics of the region, but that would be an entirely different study.

Without this central information, I do not find justification for the claim that "all children" were screened or indeed for the study as it stands as the impact of selection in this population cannot be assessed.